# Be Confident: Uncovering Overfitting in MLLM Multi-Task Tuning

**Wenke Huang** [* 1]   **Jian Liang** [* 1]   **Guancheng Wan** [1]   **Didi Zhu** [2]   **He Li** [1]   **Jiawei Shao** [3]   **Mang Ye** [† 1]   **Bo Du** [† 1]
**Dacheng Tao** [4]

## Abstract

Fine-tuning Multimodal Large Language Models (MLLMs) in multi-task learning scenarios has emerged as an effective strategy for achieving cross-domain specialization. However, multi-task fine-tuning appears performance degradation on open-response datasets. We posit that free-form answer generation primarily depends on language priors, and strengthening the integration of visual behavioral cues is critical for enhancing prediction robustness. In this work, we propose Noise Resilient Confidence Alignment to address the open-response overfitting challenge during multi-task fine-tuning. Our approach prioritizes maintaining consistent prediction patterns in MLLMs across varying visual qualities. To achieve this, we synthesize distorted visual inputs and enforce token prediction confidence alignment towards normal visual branch. By explicitly linking confidence calibration to visual robustness, this method reduces over-reliance on language priors. We conduct extensive empirical evaluations across diverse multi-task downstream via popular MLLM architectures. The comprehensive experiment demonstrates our effectiveness, showcasing its ability to alleviate open-response overfitting while maintaining satisfying multi-task performance.

## 1. Introduction

Driven by the remarkable success of Large Language Model (LLM) in the natural language processing and understand-

---

[*]Equal contribution; Work done during an internship at TeleAI
[1]National Engineering Research Center for Multimedia Software, School of Computer Science, Wuhan University, Wuhan, China [2]Department of Computer Science and Technology, Zhejiang University, Hangzhou, China [3]Institute of Artificial Intelligence (TeleAI), China Telecom, China [4]Nanyang Technological University, Singapore. Correspondence to: Mang Ye <yemang@whu.edu.cn>, Bo Du <dubo@whu.edu.cn>.

*Proceedings of the $42^{nd}$ International Conference on Machine Learning*, Vancouver, Canada. PMLR 267, 2025. Copyright 2025 by the author(s).

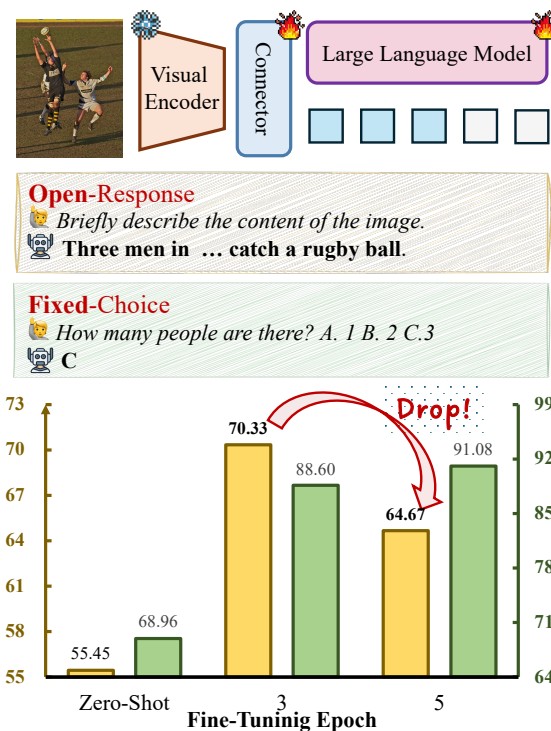

Figure 1: **Background and Problem**. Fine-tuning Multimodal Large Language Model (MLLM) on **Multi-Task** scenarios enables the model to acquire diverse specialization capabilities. This process involves training (🔥) both the connector and LLM modules for $E$ epochs. We notice that during tuning process, Open-Response task reveals a notable **performance degradation**, while Fixed-Choice task demonstrates consistent stability. Experiments are conducted on Flickr30k and ScienceQA with VILA architecture.

ing (Devlin et al., 2018; Radford et al., 2019; Brown et al., 2020; Chowdhery et al., 2022; OpenAI, 2023), Many efforts have been made to extend LLM to Multimodal Large Language Model (MLLM), which have demonstrated remarkable capabilities in generating coherent and contextually relevant descriptions from visual inputs (Liu et al., 2023b; Dai et al., 2023; Lin et al., 2023; Liu et al., 2023a). Specifically, MLLM generally follows the paradigm to fuse the pre-trained vision encoder (Radford et al., 2021; Dosovitskiy et al., 2021) into the representation space of the Large Language Model, *e.g.*, LLaMA (Touvron et al., 2023) and Vicuna (Chiang et al., 2023), via the connector module (Dai et al., 2023; Liu et al., 2023b; Luo et al., 2024). Consider-

ing that Multimodal Large Language Model is optimized on millions of multimodality instruction-following datasets (Lin et al., 2014; Singh et al., 2019; Mishra et al., 2019), it brings powerful generalization ability on different related tasks. Despite this, MLLM still performs poorly on different downstream tasks (Ren et al., 2023; Zhou et al., 2024).

Towards adapting to the target domain, the straightforward solution is to fine-tune on the task-specific dataset (Zhou et al., 2024; Han et al., 2024). With respect to existing explorations, they normally focus on **single-task** adaption, it would be simple to construct the personalized fine-tuning paradigm. However, facing the **multi-task** tuning requirement, we are curious what challenge would incur. To visualize our exploration, we conducted extensive experiments on two major downstream tasks: open-response and fixed-choice tasks. We randomly select datasets from these two views to combine the fine-tuning resource. As shown in Fig. 1, *with longer training epochs, the open-response task appears the overfitting tendency* (Guo et al., 2017; Lakshminarayanan et al., 2017) and brings the performance degradation on the corresponding testing dataset. We further investigate **why** Multimodal Large Language Model (MLLM) overfitting excites. As noted in previous studies (Favero et al., 2024; Wang et al., 2024a; Li et al., 2023b), MLLM tends to rely on the language priors to generate description sentences, consequently allocating less attention to visual information. Therefore, particularly in open-response tasks, MLLM tends to overfit to the current distribution, resulting in constrained specialization performance and an increased likelihood of fabricating content (Huang et al., 2023; Gekhman et al., 2024). Thus, a fundamental question naturally emerges: *How to alleviate open-response overfitting during multi-task MLLM fine-tuning?*

The naive solution is to establish asynchronous training epochs, *i.e.*, shorter for caption and longer for VQA. However, this operation would incur catastrophic forgetting on the caption task. In this work, we seek to mitigate overfitting in MLLM fine-tuning by examining the problem from an **inner visual behavior** perspective. Specifically, we find that when confronted with low-quality images, MLLM increasingly relies on textual modality prior information rather than visual cues (Leng et al., 2024; Favero et al., 2024; Wang et al., 2024a; Li et al., 2023b). This observation suggests that maximizing the mutual information between different visual input views during fine-tuning could help counterbalance the textual bias. Motivated by this insight, we propose Noise Resilient Confidence Alignment (NRCA), a method inspired by contrastive learning paradigms that encourage augmentation invariance and instance discrimination (Ye et al., 2019; 2020; He et al., 2020; Chen et al., 2020; Huang et al., 2021; Wu et al., 2022; Wang et al., 2024c). However, directly applying a standard contrastive paradigm in MLLM for visual coherence two challenges. **First**, common data augmentation strategies (*e.g.*, color jitter, random cropping) (Tian et al., 2020; Chen et al., 2024a; Woo et al., 2024; Kim et al., 2024b) can drift from the original answer. *To address this issue*, we propose the Noisy Visual Mixup which utilizes the Gaussian noise to construct the distorted visual signal, which does not modify the semantic level knowledge. **Second**, the inherently variable prediction length of MLLM makes straightforward contrastive regularization unstable and potentially disruptive. *To overcome this problem*, we introduce the Token Confidence Alignment, which calculates the overall prediction token confidence and expects to achieve the confidence-level alignments. The rationale behind this is that beyond local empirical minimization, we encourage the noisy visual input to bring coherent prediction confidence with the original ones to alleviate textual bias and encourage visual representation ability. For a thorough examination, we conduct experiments on multi-tasking scenarios. We mix up and fine-tune on two major downstream tasks, *i.e.*, open-response and fixed-choice. The main contributions are summarized as follows:

- We focus on the multi-task Multimodal Large Language Model fine-tuning scenario and reveal that open-response tasks, such as image captioning, exhibit performance degradation due to language prior overfitting.

- We propose Noise Resilient Confidence Alignment (NRCA), a method designed to mitigate the open-response overfitting phenomenon by enhancing visual cues during Multimodal Large Language Model tuning process. Generally speaking, we construct noisy visual inputs and encourage token confidence alignment with the normal branch to reinforce visual representation.

- We perform a comprehensive analysis on multi-task scenarios, including both open-response datasets (Flickr30k (Young et al., 2014) and COCO-Cap (Lin et al., 2014)) and fixed-choice datasets (ScienceQA (Lu et al., 2022) and IconQA(Lu et al., 2021)). Experiments are conducted on the VILA (Lin et al., 2023) and LLaVA (Liu et al., 2023b) architectures. Through a series of ablation studies, the promising results empirically validate the effectiveness of NRCA in alleviating open-response overfitting and enhancing multi-task fine-tuning overall performance.

## 2. Related Works

### 2.1. Multimodal Large Language Models

Motivated by the great progress of Large Language Model (LLM) (Radford et al., 2019; Brown et al., 2020; OpenAI, 2023; Touvron et al., 2023; Shen et al., 2024b; Jin et al., 2024; 2025), researchers have been actively exploring ways to combine pre-trained LLM and vision encoders to construct the end-to-end Multimodal Large Language Model (MLLM) capable of processing image-text. Existing solutions follow to feed the visual features produced by a pre-trained vision encoder into the LLM module with the

visual projector, *e.g.*, Flamingo (Alayrac et al., 2022), BLIP (Li et al., 2022a; 2023a), InstructBLIP (Dai et al., 2023), QWen-VL (Bai et al., 2023), LLaVA (Liu et al., 2023b;a; Zhu et al., 2024c; Li et al., 2024a; Luo et al., 2024), VILA (Lin et al., 2023; Fang et al., 2024), Cambrian-1 (Tong et al., 2024). Although MLLM brings the powerful generalization ability on a wide range of tasks (Marino et al., 2019; Lu et al., 2021; Lin et al., 2014; Kazemzadeh et al., 2014) but appears limited performance on specific domains. Thus, fine-tuning MLLM on the target task brings a reasonable solution for boosting downstream performance (Zhou et al., 2024).

### 2.2. Fine-Tune MLLM

Pre-trained MLLM on large-scale datasets can be easily transferred to downstream vision tasks through fine-tuning (Huang et al., 2025). Existing solutions can be primarily categorized into two types. **I)** Reparameterization Fine-Tuning (Hu et al., 2022; Zhang et al., 2023; Wang et al., 2023; Hao et al., 2024; Liu et al., 2024c; Lin et al., 2024; Hu et al., 2024; Bi et al., 2025a;b; Liang et al., 2025) normally apply low-rank matrices to approximate weight changes during fine-tuning and can merge with pre-trained weights prior to inference. Despite certain advantages, it faces several limitations that hinder application. First, it normally achieves sub-optimal performance compared with direct weight updates (Biderman et al., 2024; Shuttleworth et al., 2024). Second, it introduces an additional parameter module, increasing the complexity of achieving universal architectural compatibility. **II)** Partial Fine-Tuning (Li et al., 2022b; Ansell et al., 2022; Li et al., 2023c; Yu et al., 2024; Zhang et al., 2024d;a; Zhu et al., 2024a; Lu et al., 2024b) focuses on directly optimizing selected candidate elements in MLLM, such as visual connectors and LLM blocks, to align with the requirements of downstream tasks. This paradigm enjoys the architecture-agnostic property and thus largely benefits MLLM transferability. In our work, we adhere to the partial fine-tuning and further provide detailed a learnable parameter discussion in the following Sec. 4.1.

### 2.3. Overfitting Behavior in MLLM Fine-Tuning

For downstream fine-tuning, the neural networks are default optimized based on the cross entropy (De Boer et al., 2005) which has been shown to overfit to the current distribution (Platt et al., 1999; Amodei et al., 2016; Guo et al., 2017). In the context of MLLM, over-fitting on new factual knowledge encourages MLLM prone to fabricating content, amplifying the risk of hallucinations (Huang et al., 2023; Gekhman et al., 2024). Substantial research efforts have been directed towards compensating vision shortcomings and could be mainly classified into two types: Vision Enhancement (Luo et al., 2024; Shang et al., 2024; Ghosh et al., 2024; Zhao et al., 2024; Li et al., 2024a; Shi et al., 2024;

Shen et al., 2024a; Guo et al., 2024) and Update Calibration (Panigrahi et al., 2023; Chen et al., 2024b). The former solution introduces additional **external vision signals**, *e.g.*, high resolution, object grounding. However, these solutions increase the computational burden and limit architecture compatibility due to the added parameters in pre-trained models (Huo et al., 2024). In contrast, Update Calibration methods derive two main research directions. ❶ *Partial Update Mask* (Li et al., 2022b; Ansell et al., 2022; Li et al., 2023c; Yu et al., 2024; Zhang et al., 2024d;a; Zhu et al., 2024a; Lu et al., 2024b; Hui et al., 2024; Chen et al., 2024b) selects the candidate parameters group to alleviate the overfitting phenomenon. For example, Random Mask-Tuning randomly selects half of the learnable parameters and freezes the rest. ❷ *Stiff Penalty Regularization* (Kirkpatrick et al., 2017; Zenke et al., 2017; Xuhong et al., 2018; Ritter et al., 2018; Buzzega et al., 2020; Li et al., 2020; Panigrahi et al., 2023) measures the parameter importance degree to construct the parameter weight updates penalty. However, existing approaches primarily focus on single-task adaptation and indiscriminately apply anti-overfitting objectives across all training samples, failing to address the fitting condition inconsistencies in multi-task setting (Dong et al., 2023) such as open-response and fixed-choice objectives. In this work, we focus on MLLM multi-task fine-tuning and reveal that open-response tasks are particularly susceptible to overfitting due to excessive dependence on textual modalities. To mitigate this limitation, we propose leveraging **intrinsic visual behavior** patterns to strengthen cross-modal representation robustness. To address this issue, we incorporate. Specifically, we synthesize perturbed visual inputs through Gaussian noise injection and enforce confidence consistency between predictions from corrupted and pristine visual branches. This dual-branch alignment mechanism effectively reduces textual bias dominance while preserving semantic fidelity, effectively counteracting overfitting tendencies during MLLM optimization.

## 3. Methodology

### 3.1. Preliminary

Given the Multimodal Large Language Model (MLLM $\theta$) architecture, the MLLM model typically includes three modules: visual encoder $f$ (Dosovitskiy et al., 2021), LLM $g$, and the connector module $\varphi$ (Liu et al., 2023b; Dai et al., 2023; Liu et al., 2023a; Lin et al., 2023). For a query instance, the input consists of both a visual image $x^v$ and a textual instruction $x^t$. Denote the vocabulary dictionary as $V$ and language response $\mathbf{y} = \{\mathbf{y}_1, \ldots, \mathbf{y}_t, \ldots, \mathbf{y}_T\} \in \mathbb{R}^T$. Thus, $\mathbf{y}_t \in V$ and $t \in \{1, \ldots, T\}$. $T$ denotes the token length. To be precise, we extract the visual features $m^v = f(x^v)$, and then apply the trainable projection $\varphi$ to convert $m^v$ into language embedding tokens, $h^v = \varphi \cdot m^v$.

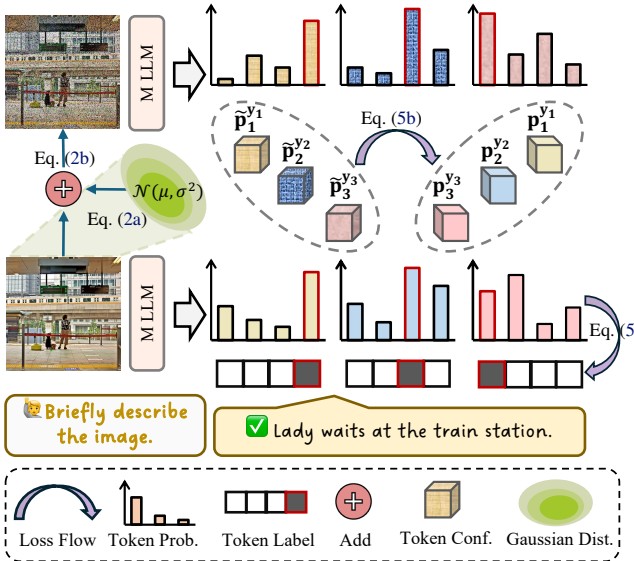

Figure 2: **Schematization of NRCA.** We propose Noisy Visual Mixup, which leverages Gaussian distribution to generate noisy visual inputs. Additionally, we introduce Token Confidence Alignment, designed to align the confidence of noisy tokens with the normal visual branch behavior, thereby enhancing visual representations during multi-task fine-tuning. The process is illustrated via logits output with the token length $T = 3$ and class number $|V| = 4$. Best viewed in color. Zoom in for details. Refer to Sec. 3.2

And textual token as $h^t = \text{Tokenize}(x^t)$. Next, we combine both visual and textual tokens and pass them into the LLM module $g$ to generate the logits output $z = g(h^v, h^t)$. In our work, following previous MLLM fine-tuning works and benchmarks (Zhou et al., 2024; Zhu et al., 2024a; Han et al., 2024) to select the trainable parameters. For the visual module, we freeze the vision encoder and tune visual connector module $\varphi$. For LLM aspect, LLM includes several transformer blocks and selects the candidate block layers set $\mathcal{N}$ for optimization as $g[L]$. Thus, we obtain learnable modules as $w = \{\varphi, g[N]\}$. Recall the fine-tuning distribution $D$, the default MLLM optimization procedure follows:

$$\arg\min_{w} \mathbb{E}_{(x^v, x^t, \mathbf{y}) \in \mathcal{D}} \mathcal{L}\left(g(h^v, h^t), \mathbf{y}\right). \quad (1)$$

The $\mathcal{L}$ normally utilizes the Cross-Entropy term (De Boer et al., 2005) for regular optimization target.

### 3.2. Noise Resilient Confidence Alignment

**Motivation**. In general, features derived from the standard visual output layer are effective for reasoning across Multimodal tasks. However, due to the language prior bias (Leng et al., 2024; Favero et al., 2024; Liu et al., 2024a; Wang et al., 2024b; Zhang et al., 2024b), Multimodal Large Language Model (MLLM), particularly during the fine-tuning stage for caption generation, tend to rely on the language-based intuition. Therefore, in our work, we focus on enhancing the visual representation to mitigate overfitting and improve

downstream performance. Inspired by the success of the contrastive paradigm (Wu et al., 2018; Ye et al., 2019; 2020; Chen et al., 2020; Khosla et al., 2020; Chen et al., 2022), we aim to encourage the MLLM to *generate consistent predictive behavior for both distorted and normal visual inputs*.

**Noisy Visual Mixup**. Compared to popular contrastive augmentation such as color jitter and random cropping (Xiao et al., 2021; Zhao et al., 2021; Wang et al., 2022; 2024c), would largely bring the hallucination prediction and fail to align with the original feedback. Thus, we model the inherent characteristics of the current image by introducing a random Gaussian distribution based on its own feature space. This approach leverages the image unique feature distribution to generate stochastic variations, capturing both its intrinsic structure and potential perturbations.

$$\mu = f_\mu(x^v), \quad \sigma = f_\sigma(x^v), \quad (2a)$$

$$\tilde{x}^v = \delta \underbrace{\mathcal{N}(\mu, \sigma^2)}_{\text{Noisy Part}} + \underbrace{(1-\delta)x^v}_{\text{Normal Part}}. \quad (2b)$$

$\mu$ and $\sigma^2$ respectively denote the mean and variance of the distribution. $\mathcal{N}$ specifies the Gaussian distribution. The $\delta$ denotes the noisy ratio and is default set as $0.5$. Then, we feed both normal $x^v$ and noisy $\tilde{x}^v$ visual information with the corresponding text prompt $x^t$ into the MLLM and respectively obtain the contextual word prediction logits as $z = g(h^v, h^t)$ and $\tilde{z} = g(\tilde{h}^v, h^t)$.

**Token Confidence Alignment**. Compared with traditional classification tasks, the prediction $z \in \mathbb{R}^{T \times c}$ for MLLM appears high output dimensions attributed to vocabulary scale, *e.g.*, $c = 32000$ in LLaMA, As a result, directly conducting distribution alignment faces challenges related to the imbalance between head-tail knowledge variations (Gu et al., 2024; 2025; Kim et al., 2024a; Ko et al., 2024), and fails to adequately reflect the reliability of the ground-truth signal. Therefore, in this work, we propose aligning token confidence rather than token distribution to mitigate overfitting during the fine-tuning stage of MLLM. Specifically, we first measure the prediction probability for the token $t$ via the following formulation:

$$\mathbf{p}_t = \sigma(z_t), \quad \tilde{\mathbf{p}}_t = \sigma(\tilde{z}_t) \in \mathbb{R}^c,$$
$$\mathbf{p} = [\mathbf{p}_t]_{t=1}^T, \quad \tilde{\mathbf{p}} = [\tilde{\mathbf{p}}_t]_{t=1}^T \in \mathbb{R}^{T \times C}. \quad (3)$$

$\sigma$ means the `softmax` function. Then, we further obtain the overall token prediction confidence based on the ground-truth labels $\mathbf{y} = \{\mathbf{y}_t\}_{t=1}^T$ as:

$$\mathcal{I} = \frac{1}{T} \sum_{t=1}^T \mathbf{p}_t^{\mathbf{y}_t}, \quad \tilde{\mathcal{I}} = \frac{1}{T} \sum_{t=1}^T \tilde{\mathbf{p}}_t^{\mathbf{y}_t}. \quad (4)$$

Naturally, we encourage the noisy prediction confidence $\tilde{\mathcal{I}}$ to align with the normal ones $\mathcal{I}$. Furthermore, applying the same penalty strength across all samples fails to account for sample-specific fitting conditions. To address this, we introduce the empirical loss as a guiding factor, allowing for tailored penalty allocation for each query sample. We introduce the following regularization term.

$$\mathcal{L}_{CE} = \frac{1}{T} \sum_{t=1}^{T} -\mathbf{1}_{\mathbf{y}_t} \log(\mathbf{p}_t), \tag{5a}$$

$$\mathcal{L}_{NRCA} = |1 - \frac{\tilde{\mathcal{I}}}{\mathcal{I}}| \times \boxed{\mathcal{L}_{CE}} . \tag{5b}$$

where $\mathbf{1}$ denotes the one-hot encoding of $\mathbf{y}_t$. We detach the normal branch signals in Eq. (5b) to avoid the MLLM optimization degradation. Empirically, we identify open-response samples based on their label length, applying this operation separately to these samples. Specifically, we observe that fixed-choice sample labels typically consist of a single word followed by the [EOS] token, with the label length as 2. Finally, we carry out the following optimization objective in fine-tuning phase:

$$\mathcal{L} = \mathcal{L}_{CE} + \lambda \mathcal{L}_{NRCA}. \tag{6}$$

$\lambda$ represents the penalization hyper-parameter that controls the strength of confidence alignment. We set $\lambda = 2$ and provide the corresponding ablation analysis in Sec. 4.2. Overall, the MLLM model is regularized to preserve the downstream distribution while promoting invariance in distorted confidence. As a result, our method effectively mitigates downstream open-response overfitting by enhancing visual behavior alignment. We provide a algorithm description in Algorithm 1 and the methodological framework in Fig. 2.

### 3.3. Discussion and Limitation

**Gaussian Noise Meets MLLM**. Existing MLLM research often employs Gaussian noise (Goodman, 1963; Tüske et al., 2015; Variani et al., 2015; Hayashi & Uchida, 2019; Ardizzone et al., 2020) to generate low-quality visual inputs, thereby amplifying language priors (Xiao et al., 2024). A common approach to mitigating MLLM hallucinations is to adjust next-token logits in a contrastive manner using the normal prediction distribution (Leng et al., 2024; Woo et al., 2024; Zhu et al., 2024b; Chen et al., 2024a; Huo et al., 2024; Zhang et al., 2024b; Xiao et al., 2024; Liu et al., 2024b). The rationale behind this approach is that disturbed inputs significantly exacerbate hallucinations, while contrastive decoding mitigates this effect by subtracting hallucinated concepts from the original distribution, thereby reducing confusion. Consequently, existing methodologies often treat noisy visual behavior as an optimization trap and disregard the corresponding feedback. In contrast, our work introduces a novel MLLM multi-task tuning approach to address open-response overfitting by encouraging prediction confidence alignment between normal and noisy visual signals. This strategy compels the network to fully leverage its visual processing capabilities, achieving consistent prediction confidence even when faced with degraded visual inputs.

**Conceptual Difference**. Fine-tuning the MLLM for downstream tasks is a straightforward approach to developing

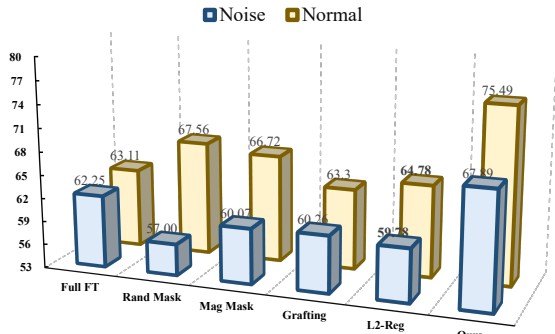

Figure 3: **Noise Evaluation Comparison**. We evaluate noisy and normal samples on Flickr30k using LLaVA with $E = 3$. Normal Full Fine-Tuning shows a limited performance gap between noisy and normal visual signals, revealing a tendency of language prior reliance. In contrast, our approach demonstrates a notable performance improvement by emphasizing the importance of visual information. See Sec. 3.3 for details.

Table 1: **Attention Allocation and Performance Comparison**. We conduct the evaluation on the Flickr30k with noisy visual input. We tuning on the Flickr30k+ScienceQA datasets based on LLaVA architecture. Please see Sec. 4.2 for detailed discussion.

| Metrics | Full FT | Ran Mask | Mag Mask | Grafting | L2-Reg | Ours |
|---|---|---|---|---|---|---|
| System (All) | 0.6837 | 0.6966 | 0.6927 | 0.6673 | 0.6791 | 0.6677 |
| Prompt (All) | 0.1229 | 0.1297 | 0.1280 | 0.1241 | 0.1181 | 0.1225 |
| Visual (All) | 0.1933 | 0.1737 | 0.1793 | 0.2084 | 0.2029 | **0.2098** |
| System (Mid) | 0.5865 | 0.5874 | 0.5859 | 0.5726 | 0.5826 | 0.5574 |
| Prompt (Mid) | 0.1733 | 0.1843 | 0.1832 | 0.1664 | 0.1623 | 0.1576 |
| Visual (Mid) | 0.2402 | 0.2282 | 0.2309 | 0.2609 | 0.2552 | **0.2850** |

domain-specialized experts. However, the limited size of downstream datasets poses a significant challenge, often leading to overfitting on target-specific behaviors and catastrophic forgetting of pre-trained knowledge. Existing studies predominantly focus on mitigating catastrophic forgetting in MLLM, where fine-tuning risks compromise the generality achieved during pre-training, thus causing upstream performance degradation. Consequently, prior research has primarily aimed to balance generalization and specialization, ensuring robust performance on both seen and unseen tasks. In contrast, our work addresses a more practical scenario by fine-tuning MLLM for multi-task applications. We argue that leveraging MLLM to achieve multi-task specialization is a more efficient approach than the conventional one-to-one fine-tuning paradigm. However, when dealing with multi-task distributions, we identify a critical issue: *MLLM frequently experiences performance degradation in open-response tasks*, *e.g.*, image caption understanding, as shown in Fig. 1. We hypothesize that this degradation arises because open-response tasks tend to rely excessively on language priors rather than leveraging visual information, as confirmed in Fig. 3. Full Fine-Tuning brings limited performance gap towards the input visual quality. Furthermore, as for Multimodal Large Language Model (MLLM), input tokens include System Tokens, Prompt Tokens and Visual

Table 2: **Incremental Resource Cost with Overall Performance Comparison**. Accuracies are derived from Flickr30k & ScienceQA based on VILA for $E = 3$. $\mathcal{O}$ denotes the complexity degree. - indicates unchanged values. ↑ means improved accuracy compared with Full FT. Please refer to Sec. 3.3.

| Metrics | Full FT | L2-Reg | Ran Mask | Our NRCA |
|---|---|---|---|---|
| Resource Cost | – | $\mathcal{O}(|\theta|)$ | $\mathcal{O}(|\theta|)$ | $\mathcal{O}(B \times (|\tilde{x}^v| + |\tilde{z}|)$ |
| Performance | 79.47 | 79.52 | 83.02 | **83.24**$_{↑3.77}$ |

Table 3: **Ablative Experiments for** $\mathcal{L}_{NRCA}$ **Formulation** in Eq. (5b). Please see Sec. 4.2 for detailed discussion.

| $\|1 - \frac{\tilde{z}}{z}\|$ | $\mathcal{L}_{CE}$ | Flickr30k | SQA | AVG | COCO-Cap | SQA | AVG |
|---|---|---|---|---|---|---|---|
| *Fine-Tune with VILA architecture* | | | | | | | |
| Zero-shot | | 55.45 | 68.96 | 62.20 | 72.57 | 68.96 | 70.76 |
| ✗ (Full FT) | | 70.33 | 88.60 | 79.47 | 109.58 | 90.33 | 99.96 |
| ✓ | | 70.87 | 90.43 | 80.65 | 114.83 | 89.59 | 102.21 |
| ✓ | ✓ | 75.10 | 90.23 | **82.67** | 120.05 | 89.94 | **105.00** |
| *Fine-Tune with LLaVA architecture* | | | | | | | |
| Zero-shot | | 25.31 | 69.56 | 47.44 | 40.28 | 69.56 | 54.92 |
| ✗ (Full FT) | | 63.11 | 88.3 | 75.71 | 101.76 | 88.65 | 95.21 |
| ✓ | | 66.82 | 87.56 | 77.19 | 104.39 | 87.80 | 96.10 |
| ✓ | ✓ | 72.39 | 86.61 | **79.50** | 114.54 | 86.32 | **100.43** |

Tokens and appear different contribution for the prediction output. Thus, we utilize the attention map between the first output token and the input token under the noisy visual input to visualize the contribution of different input token. Besides, recent works (Che et al., 2025; Zhang et al., 2025; 2024c) have shown that the MLLM tends to extract object information from the image at the middle layers. Thus, we plot the all layers and middle layers attention allocation for the first output token in the following Tab. 1. The results reveal that our method allocates more attention weights on the visual cue. Therefore, both performance comparisons and attention analyses show that our method reduces reliance on language priors and improves overall model performance by encouraging consistent predictions, even under distorted visual inputs—highlighting the value of posterior guidance over prior dependence.

**Limitations**. Our method, NRCA constructs additional noisy visual inputs to obtain the corresponding prediction confidence and expects confidence alignment with the normal branch. As a result, our method incurs extra storage requirements for the noisy input $\tilde{x}^v$ and the logits output signals $\tilde{z}$. However, existing methods typically require storing the full pre-trained parameters to apply parameter stiffness restrictions or sparse mask updating strategies, which heavily depend on the MLLM model scale ($\mathcal{O}(|\theta|)$). As shown in Table 2, while NRCA introduces an incremental storage complexity as $\mathcal{O}(B \times (|\tilde{x}^v| + |\tilde{z}|))$, this cost is relatively small compared to existing methods. Additionally, NRCA achieves the highest performance of 83.24, demonstrating that it effectively balances resource efficiency and downstream performance. Detailed performance comparison

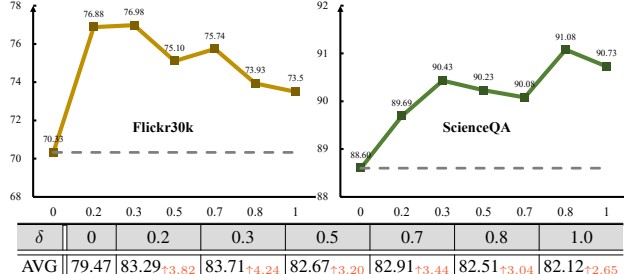

| $\delta$ | 0 | 0.2 | 0.3 | 0.5 | 0.7 | 0.8 | 1.0 |
|---|---|---|---|---|---|---|---|
| AVG | 79.47 | 83.29$_{↑3.82}$ | 83.71$_{↑4.24}$ | 82.67$_{↑3.20}$ | 82.91$_{↑3.44}$ | 82.51$_{↑3.04}$ | 82.12$_{↑2.65}$ |

Figure 4: **Ablation on Noisy Ratio** $\delta$ in Eq. (2b) effect for **image captioning** (Left), **visual question-answering** (Middle), and overall performance (Right) with VILA. $\delta = 0$ degrades into the Full Fine-Tuning. ↑ means improved accuracy compared with Full FT. Refer to Sec. 4.2 for the explanation.

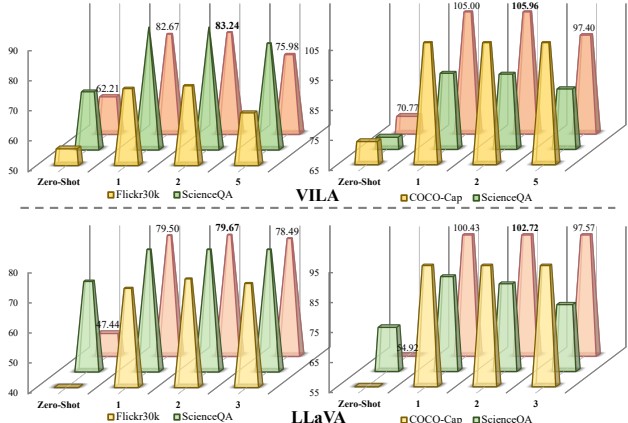

Figure 5: **Ablation on** the hyper-parameter $\lambda$ in Eq. (6) for the proposed method with different MLLM architectures. We default set $\lambda = 2$ in the following experiments. See Sec. 4.2. for details.

refers to Sec. 4.3. Furthermore, when the downstream distribution consists solely of fixed-choice tasks, MLLM does not encounter significant performance degradation. However, open-response tasks are crucial for unlocking MLLM ability to generate free-form expressions, which go beyond the limitations of traditional classification. In the context of multi-task settings, we argue that open-response tasks play an essential role in showcasing the full potential of dialogue experience of MLLM, even though they introduce challenges such as performance drops due to overfitting on language information priors.

## 4. Experiments

### 4.1. Experimental Setup

**Architecture and Datasets**. Adhering to the Multimodal Large Language Model (MLLM) paradigm, we evaluate the effectiveness of our proposed methods using two widely adopted models, LLaVA(Liu et al., 2023b) and VILA(Lin et al., 2023), as the foundational MLLM for our experiments. We categorize the four downstream datasets into two task types: **image captioning** (open-response) and **visual**

Table 4: **Comparison with the state-of-the-art Multimodal Large Language Model (MLLM) Fine-Tuning Solutions** on the multi-task setting: **image captioning** (Flickr30k and COCO-Cap) and **visual question-answering** (ScienceQA and IconQA) based on VILA and LLaVA architectures with tuning epoch $E = 3$. We mark the Best in bold and Second in underline across different tuning methods. $+$ means improved accuracy compared with Zero-shot. Please refer to Sec. 4.3 for relative explanations.

| Methods | Flickr30k | ScienceQA | AVG | COCO-Cap | ScienceQA | AVG | Flickr30k | IconQA | AVG | COCO-Cap | IconQA | AVG |
|---|---|---|---|---|---|---|---|---|---|---|---|---|
| *Fine-Tune with VILA architecture* | | | | | | | | | | | | |
| Zero-shot | 55.45 | 68.96 | 62.20 | 72.57 | 68.96 | 70.76 | 55.45 | 55.57 | 55.51 | 72.57 | 55.57 | 64.07 |
| Full FT | 70.33 | 88.60 | 79.47 | 109.58 | 90.33 | 99.96 | 68.20 | 86.62 | 77.41 | 108.16 | 86.91 | 97.53 |
| Ran Mask | 74.77 | 91.27 | 83.02 | 113.86 | 90.68 | 102.27 | 73.16 | 86.57 | 79.87 | 113.59 | 86.64 | 100.12 |
| Mag Mask | 75.37 | 91.08 | 83.23 | 114.12 | 90.28 | 102.20 | 73.13 | 87.32 | 80.23 | 113.29 | 87.11 | 100.20 |
| Grafting | 70.23 | 89.14 | 79.69 | 109.02 | 90.93 | 99.98 | 69.01 | 86.67 | 77.84 | 107.73 | 87.44 | 97.59 |
| L2-Reg | 68.71 | 90.33 | 79.52 | 108.94 | 90.48 | 99.71 | 69.3 | 87.22 | 78.26 | 108.69 | 87.00 | 97.85 |
| NRCA | 76.00 | 90.48 | **83.24** | 122.27 | 89.64 | **105.96** | 75.98 | 86.99 | **81.49** | 120.82 | 86.19 | **103.51** |
| | + 20.55 | + 21.52 | + 21.04 | + 49.70 | + 20.68 | + 35.20 | + 20.53 | + 31.42 | + 25.98 | + 48.25 | + 30.62 | + 39.44 |
| *Fine-Tune with LLaVA architecture* | | | | | | | | | | | | |
| Zero-shot | 25.31 | 69.56 | 47.44 | 40.28 | 69.56 | 54.92 | 25.31 | 48.42 | 36.87 | 40.28 | 48.42 | 44.35 |
| Full FT | 63.11 | 88.30 | 75.71 | 101.76 | 88.65 | 95.21 | 64.14 | 82.68 | 73.41 | 101.68 | 83.06 | 92.37 |
| Ran Mask | 67.56 | 85.62 | 76.59 | 105.24 | 85.82 | 95.53 | 66.64 | 79.65 | 73.15 | 104.86 | 79.69 | 92.28 |
| Mag Mask | 66.72 | 85.72 | 76.22 | 105.67 | 85.57 | 95.62 | 66.70 | 79.54 | 73.12 | 105.94 | 79.56 | 92.75 |
| Grafting | 63.30 | 89.44 | 76.37 | 100.47 | 89.64 | 95.06 | 66.35 | 83.91 | 75.13 | 101.20 | 83.76 | 92.48 |
| L2-Reg | 64.78 | 90.03 | 77.41 | 101.05 | 89.94 | 95.50 | 65.24 | 83.95 | 74.60 | 101.92 | 83.72 | 92.82 |
| NRCA | 75.49 | 83.84 | **79.67** | 121.50 | 83.94 | **102.72** | 74.31 | 77.60 | **75.96** | 118.74 | 77.83 | **98.29** |
| | + 50.18 | + 14.28 | + 32.23 | + 81.22 | + 14.38 | + 47.80 | + 49.00 | + 29.18 | + 39.09 | + 78.46 | + 29.41 | + 53.94 |

**question-answering** (fixed-choice), as follows:

- Flickr30k (Young et al., 2014): Design for understanding the visual content of images associated with linguistic expressions, it contains $31,000$ images from Flickr, each paired with 5 human-annotated reference sentences.
- COCO-Cap (Lin et al., 2014): Comprise over $330,000$ images with more than 1.5 million captions, this dataset provides 5 human-generated captions per image for both training and validation.
- ScienceQA (Lu et al., 2022): Collect from elementary and high school science curricula. This dataset includes approximately $21,000$ multimodal multiple-choice questions spanning diverse science topics.
- IconQA (Lu et al., 2021): Focus on abstract diagram understanding and cognitive reasoning in real diagram.

Specifically, we follow the training settings from prior works (Zhou et al., 2024; Zhu et al., 2024a) and utilize official optimization scripts[1]. For fixed-choice tasks (ScienceQA and IconQA), we use the textual prompt: *"Answer with the option's letter from the given choices directly."* For open-response tasks (Flickr30k and COCO-Cap), we collect five manually written instructions and randomly sample one as the prompt for each caption, as follows:

- *"Describe the image as simply as possible with a sentence or phrase"*
- *"Give a brief summary of what you see"*
- *"Provide a short description of the image"*
- *"Write a short description for the image"*
- *"Briefly describe the content of the image"*

---

[1] https://huggingface.co/datasets/BAAI/DataOptim

For each dataset, we randomly sample $10k$ instances from the training set. We randomly select datasets from the above mentioned captioning and VQA tasks to construct the multi-task training datasets.

**Counterparts**. We focus on exploring anti-overfitting MLLM fine-tuning methods and mainly compare with the ❶ *Partial Update Mask* and ❷ *Stiff Penalty Regularization* solutions as the following formulations:

- Full Fine-Tuning (Full FT) [arXiv'05] (De Boer et al., 2005): Default optimize full candidate parameters towards the downstream distribution.
- L2-Regularization (L2-Reg) [PNAS'17] (Kirkpatrick et al., 2017): Add $\mathcal{L}_2$ regularization term with the regularization hyper-parameter, *i.e.*, 1e-3, to the original loss function. It focuses on keeping the fine-tuning model closer to the pre-trained model, thereby mitigating forgetting.
- Grafting [ICML'23] (Panigrahi et al., 2023): Localize newly acquired skills inside fine-tuned language models, which could be regarded as $\mathcal{L}_1$ regularization with the penalty weigh, *i.e.*, 1e-6.
- Random Mask-Tuning (Ran Mask) [arXiv'24] (Hui et al., 2024): Randomly update half ratio of parameters within each transformer layer at each iteration while freezing the other elements. We default set the sampling ratio as $50\%$.
- Magnitude Tuning (Mag Mask) [NeurIPS'15] (Han et al., 2015)): Select and maintain elements with relative large weight magnitude with the pre-defined sampling ratio. The default updating proportion is set as $50\%$.

**Implementation Details**. We follow the official reposito-

Table 5: **Comparison with the state-of-the-art Multimodal Large Language Model (MLLM) Fine-Tuning Solutions** on the multi-task setting: **image captioning** (Flickr30k and COCO-Cap) and **visual question-answering** (ScienceQA and IconQA) based on VILA and LLaVA architectures with tuning epoch $E = 5$. We mark the Best in bold and Second in underline across different tuning methods. $+$ means improved accuracy compared with Zero-shot. Please refer to Sec. 4.3 for relative explanations.

| Methods | Flickr30k | ScienceQA | AVG | COCO-Cap | ScienceQA | AVG | Flickr30k | IconQA | AVG | COCO-Cap | IconQA | AVG |
|---|---|---|---|---|---|---|---|---|---|---|---|---|
| *Fine-Tune with VILA architecture* | | | | | | | | | | | | |
| Zero-shot | 55.45 | 68.96 | 62.20 | 72.57 | 68.96 | 70.76 | 55.45 | 55.57 | 55.51 | 72.57 | 55.57 | 64.07 |
| Full FT | 64.67 | 91.08 | 77.88 | 102.23 | 91.22 | 96.73 | 65.42 | 87.68 | 76.55 | 100.40 | 87.29 | 93.85 |
| Ran Mask | 66.50 | 90.98 | 78.74 | 106.52 | 90.98 | 98.75 | 67.43 | 86.95 | 77.19 | 105.4 | 87.21 | 96.31 |
| Mag Mask | 69.17 | 91.77 | 80.47 | 105.16 | 91.22 | 98.19 | 67.34 | 87.24 | 77.29 | 104.98 | 86.87 | 95.92 |
| Grafting | 64.76 | 91.03 | 77.90 | 100.16 | 90.13 | 95.15 | 62.4 | 87.82 | 75.11 | 99.43 | 87.43 | 93.43 |
| L2-Reg | 63.13 | 91.22 | 77.18 | 101.98 | 91.67 | 96.83 | 61.72 | 87.86 | 74.79 | 100.34 | 87.46 | 93.90 |
| NRCA | 70.87 | 90.58 | **80.73** | 117.54 | 90.78 | **104.16** | 74.95 | 87.22 | **81.08** | 117.99 | 86.92 | **102.46** |
| | +15.42 | +21.62 | +18.53 | +44.97 | +21.82 | +33.40 | +19.50 | +31.65 | +25.57 | +45.42 | +31.35 | +38.39 |
| *Fine-Tune with LLaVA architecture* | | | | | | | | | | | | |
| Zero-shot | 25.31 | 69.56 | 47.44 | 40.28 | 69.56 | 54.92 | 25.31 | 48.42 | 36.87 | 40.28 | 48.42 | 44.35 |
| Full FT | 61.46 | 91.08 | 76.27 | 92.82 | 90.88 | 91.85 | 60.51 | 84.69 | 72.60 | 92.19 | 84.48 | 88.34 |
| Ran Mask | 62.31 | 88.84 | 75.58 | 95.91 | 88.30 | 92.11 | 62.14 | 82.66 | 72.40 | 91.98 | 81.62 | 86.80 |
| Mag Mask | 63.10 | 88.55 | 75.83 | 96.88 | 88.50 | 92.69 | 61.37 | 82.43 | 71.90 | 95.23 | 82.79 | 89.01 |
| Grafting | 61.40 | 91.82 | 76.61 | 92.91 | 91.92 | 92.42 | 101.2 | 83.76 | 92.48 | 90.84 | 85.16 | 88.00 |
| L2-Reg | 59.77 | 91.52 | 75.65 | 91.76 | 91.67 | 91.72 | 59.18 | 85.99 | 72.59 | 91.52 | 85.32 | 88.42 |
| NRCA | 73.04 | 87.11 | **80.08** | 120.81 | 82.80 | **101.81** | 74.51 | 81.65 | **78.08** | 120.31 | 78.34 | **99.33** |
| | +47.73 | +17.55 | +32.64 | +80.53 | +13.24 | +46.89 | +49.20 | +33.23 | +41.21 | +80.03 | +29.92 | +54.98 |

ries[2,3] to conduct the fine-tuning procedure. With respect to the training scenarios, we first randomly sample $10,000$ instances from each dataset and combine them as multi-task tuning datasets. As for the optimization details, the learning rate $lr$ in LLaVA (Liu et al., 2023b) is $2e-4$ for LLM and $2e-5$ for the visual projector. For VILA(Lin et al., 2023), we uniformly set the learning rate to $1e-4$. The training epochs are set to $E = 3$ and $E = 5$. The training batch size $B$ is set to 16 by default. The fine-tuning block for LLM is the *last $N = 2$ layers*. Regarding the experimental conditions, all experiments are conducted on 8 NVIDIA 4090 GPUs, each with 24GB of memory. Due to the limited computational resources, we select the LLaVA-1.5-7B for LLaVA and VILA1.5-3B for VILA.

**Evaluation Metrics**. To evaluate the performance of MLLM across different downstream task types, we utilize CIDER (Vedantam et al., 2015) and Top-1 Accuracy metrics for image captioning and visual question-answering tasks, respectively. Additionally, we compute the mean results to represent the overall downstream performance.

### 4.2. Diagnostic Analysis

We perform ablation studies on two scenarios: Flickr30k & ScienceQA and COCO-Cap & ScienceQA, utilizing both the VILA and LLaVA architectures with a tuning epoch of $E = 3$ to facilitate an in-depth analysis.

**Proposed Training Objective**. We quantitatively analyze the proposed Noise Resilient Confidence Alignment

(NRCA) in Tab. 3. The ablation results demonstrate that directly encouraging prediction behavior consistency effectively mitigates the open-response overfitting phenomenon. Moreover, incorporating empirical loss guidance, which accounts for sample difficulty, further enhances the overall performance. Thus, combining Noisy Visual Mixup and Token Confidence Alignment acquires satisfying downstream multi-task performance that coincides with our motivation of exploiting the noisy resilience to alleviate the open-response overfitting during Multimodal Large Language Model tuning.

**Noisy MixUp Ratio**. The parameter $\delta$, introduced in Eq. (2b), controls the Gaussian distortion ratio. As illustrated in Fig. 4, increasing the distortion ratio poses challenges for confidence alignment in open-response tasks, resulting in limited performance improvements. Generally, as $\delta$ increases, a trade-off between open-response and fixed-choice performance becomes more pronounced, with improvements diminishing under higher noise ratios. For consistency and convenience, we set $\delta = 0.5$ across different scenarios in the subsequent experiments.

**Control Hyper-Parameter $\lambda$ in Eq. (6)**. The Sec. 4 quantifies the effect of hyper-parameter $\lambda$, which measures the strength of token confidence difference penalization on different scenarios. Specifically, the open-response and fixed-task trade-off performance progressively mounts as $\lambda$ enlarges, and the improvement presents marginal under strict parameter stiffness. For convenience, we choose the $\lambda = 2$ for different scenarios in the following experiments.

[2]https://github.com/haotian-liu/LLaVA
[3]https://github.com/NVlabs/VILA

### 4.3. Comparison to State-of-the-Arts

We compare our NRCA against related approaches on the multi-task scenarios: image-captioning & visual question-answering tasks. As shown in Tab. 4 and Tab. 5, several key observations can be made. **First**, *Partial Update Mask* does not fundamentally mitigate overfitting but merely slows the fitting process of MLLM. This is evident from its limited performance gains, even with extended tuning epochs $E$ and larger model architectures, *e.g.*, scaling from 3B to 7B. Specifically, Random Mask-Tuning applies stochastic masks to parameters, injecting noise to reduce reliance on specific sub-networks and implicitly regularize by limiting parameter co-adaptation. While effective on smaller models (VILA-3B), its performance drops on larger ones (LLaVA-7B), suggesting its effect comes from constrained optimization rather than addressing language prior dominance. **Second**, *Stiff Penalty Regularization* is highly sensitive to regularization strength, making it difficult to maintain stable performance improvements across varying settings with consistent hyper-parameters. For instance, in VILA with COCO-Cap and IconQA under $E = 5$, Grafting demonstrates inferior performance compared to Full Fine-Tuning, highlighting its limitations in achieving robust and consistent results. **Third**, our method NRCA promotes prediction confidence alignment and consistently delivers robust results across various downstream tasks and model architectures. Moreover, as demonstrated in Tab. 2, our approach requires fewer computational resources compared to existing methods, highlighting its efficiency.

## 5. Conclusion

In our work, we focus on fine-tuning Multimodal Large Language Model (MLLM) in multi-task scenarios to achieve multi-domain specialization. However, we observe a critical issue: open-response distributions appear performance degradation during the training process. We attribute this degradation to over language prior reliance. To address this, we propose Noise Resilient Confidence Alignment (NRCA), which aims to enhance the effect of visual cues in the prediction process. Generally speaking, we leverage Gaussian distortions to construct noisy visual inputs and encourage overall token confidence alignment with the normal visual branch behavior. Compared to existing methods, our approach offers two key advantages: First, *Resource Cost Decline*: NRCA conducts output calibration without requiring storage of pre-trained Multimodal Large Language Model weights to counteract overfitting phenomenon, making ours computationally efficient compared to existing methods. Second, *Hyper-Parameter Stability*: Related methods often rely on pre-defined mask ratios or decorate parameter stiffness control, which are highly correlated with the architecture and scale of MLLM. In contrast, our method

introduces prediction confidence calibration that shows the model-agnostic property. NRCA has been validated across various scenarios, demonstrating its effectiveness and highlighting its potential for broader applications.

## Acknowledgement

This work is supported by National Natural Science Foundation of China under Grant (62225113, 62361166629, 62176188, 623B2080), the National Key Research and Development Program of China (2023YFC2705700, 2024YFC3308400), and the Wuhan University Undergraduate Innovation Research Fund Project. The supercomputing system at the Supercomputing Center of Wuhan University supported the numerical calculations in this paper. Dr Tao's research is supported by NTU RSR and Start Up Grants.

## Impact Statement

This paper presents work whose goal is to advance the field of Machine Learning. There are many potential societal consequences of our work, none of which we feel must be specifically highlighted here.

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

# APPENDIX

## A. Notation Table

We provide the notation table in Tab. 6.

Table 6: **Notations** table.

| | Description | | Description |
|---|---|---|---|
| $\theta$ | MLLM | $f$ | Vision encoder |
| $\varphi$ | Connector | $g$ | Large Language Model |
| $\mathbf{y}$ | Language response | $T$ | Label Length |
| $V$ | Vocabulary dictionary | $N$ | LLM tuning blocks |
| $x^v$ | Visual image | $\mathcal{N}$ | Gaussian distribution |
| $\tilde{x}^v$ | Noisy visual image | $x^t$ | Textual instruction |
| $m^v$ | Visual feature | $h^v$ | Visual token embedding |
| $h^t$ | Text token embedding | $z$ | Logits output |
| $\mathbf{p}_t$ | Token Probability | $\mathbf{p}_t^{\mathbf{y}_t}$ | Token Confidence |
| $\mathcal{I}$ | Overall token prediction | $\sigma$ | `softmax` function |
| $E$ | Fine-Tuning epoch | $D$ | Downstream dataset |
| $\mathcal{L}$ | Loss function | $\lambda$ | Control hyper-parameter |
| $\eta$ | Learning rate | $B$ | Training batch size |

## B. Algorithm

We provide the algorithm description in Algorithm 1.

---

**Algorithm 1** NRCA

---

**Input:** Fine-Tuning epoch $E$, Downstream dataset $D$
Overall MLLM Network $\theta$, Trainable parameter module $w$
**Output:** The optimized selected MLLM module $w$

**for** $e = 1, 2, ..., E$ **do**
  **for** $(x^v, x^t, \mathbf{y}) \in D$ **do**
    /* Construct Noisy Vision View */
    $\mu, \sigma \leftarrow (x^v)$ via Eq. (2a)
    $\tilde{x}^v \leftarrow (x^v, \mu, \sigma)$ by Eq. (2b)

    $h^v = \varphi(f(x^v))$ and $\tilde{h}^v = \varphi(f(\tilde{x}^v))$
    $h^t = \text{Tokenize}(x^t)$
    $z = g(h^v, h^t)$ and $\tilde{z} = g(\tilde{h}^v, h^t)$

    /* Token Confidence Alignment */
    $\mathbf{p}_t = \sigma(z_t), \tilde{\mathbf{p}}_t = \sigma(\tilde{z}_t)$ ;     // Token Prob.
    $\mathbf{p} = [\mathbf{p}_t^{\mathbf{y}_t}]_{t=1}^T, \tilde{\mathbf{p}} = [\tilde{\mathbf{p}}_t^{\mathbf{y}_t}]_{t=1}^T$ ;   // Token Conf.
    $\mathcal{L}_{NRCA} \leftarrow (\mathbf{p}, \tilde{\mathbf{p}})$ through Eq. (5b)

    $\mathcal{L}_{CE} \leftarrow (\mathbf{p}, \mathbf{y})$ in Eq. (5a)
    $\mathcal{L} = \mathcal{L}_{CE} + \lambda \mathcal{L}_{NRCA}$

    $w = w - \eta \nabla \mathcal{L}$ ;     // Update Param.
  **end**
**end**

---

## C. Rationale Analysis

We conduct a theoretical discussion to investigate the rationale for regularizing the prediction confidence in the Multimodal Large Language Model (MLLM) to mitigate

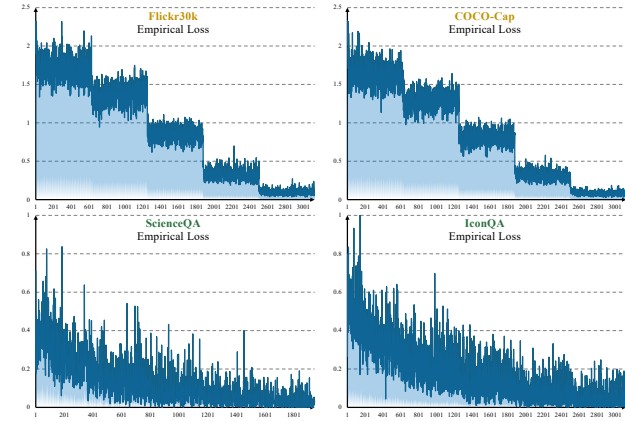

Figure 6: **Empirical Loss** on both **Open-Response** and **Fixed-Choice** tasks. Experiments are based on the LLaVA with tuning epoch $E = 5$. We observe a *stepwise loss* tendency in the former. The open-response distribution compels MLLM to over-memorize each sample during every training epoch, resulting in a jumping loss behavior. Please refer to Appendix C.

overfitting. As a preliminary step, we revisit the classical MLLM fine-tuning optimization loss. Specifically, we suppose $(x^v, x^t, \mathbf{y})$ as the query instance from the training dataset, where $\mathbf{y} \in \mathbb{R}^{T \times |V|}$ is the ground-truth label. We obtain the prediction token logits as $z = \theta(x^v, x^t)$. Then, we formulate the cross-entropy (De Boer et al., 2005) as:

$$
\begin{aligned}
\mathcal{L}_{CE} &= \frac{1}{T} \sum_{t=1}^T -\mathbf{1}_{\mathbf{y}_t} \log(\sigma(z)), \\
&= \frac{1}{T} \sum_{t=1}^T -\mathbf{1}_{\mathbf{y}_t} \log(\mathbf{p}_t), \qquad (7) \\
&= \frac{1}{T} \sum_{t=1}^T \log(\mathbf{p}_t^{\mathbf{y}_t}),
\end{aligned}
$$

where $\mathbf{1}_{\mathbf{y}_t}$ denotes the one-hot encoding of $\mathbf{y}_t$, and $\sigma$ represents the `softmax` operation (Wang et al., 2017). Thus, to some extent, minimizing $\mathcal{L}_{CE}$ can be interpreted as maximizing the prediction confidence for each token, $\mathbf{p}_t^{\mathbf{y}_t}$.

Consequently, MLLM, with its large-scale network capacity, often encounters less challenging scenarios when handling downstream tasks with limited scope, making it prone to memorizing the training samples. In open-response tasks, a long ground-truth label sequence is typically involved. As a result, during empirical loss minimization, each token prediction confidence naturally increases to align with the target distribution. In contrast, for fixed-choice tasks, where only a single ground-truth token label usually exists (*i.e.*, $T = 1$), responses are constrained to predefined options such as A, B, C, or D. This constrained candidate label space makes it more challenging to enforce memorization of each individual sample.

Therefore, we respectively plot the training loss curve $\mathcal{L}_{CE}$ for both **open-response** and **fixed-choice** tasks. As shown

Table 7: **Comparison with the state-of-the-art Multimodal Large Language Model (MLLM) Fine-Tuning Solutions** on the multi-task setting: **image captioning** (Flickr30k and COCO-Cap) and **visual question-answering** (ScienceQA and IconQA) based on the VILA architectures with tuning epoch $E = 3$. We mark the Best in bold and Second in underline across different tuning methods. $+$ means improved accuracy compared with Zero-shot. Please refer to Appendix D for relative explanations.

| Methods | Flickr30k | ScienceQA | AVG | COCO-Cap | ScienceQA | AVG | Flickr30k | IconQA | AVG | COCO-Cap | IconQA | AVG |
|---|---|---|---|---|---|---|---|---|---|---|---|---|
| *Fine-Tune with $B = 32$* | | | | | | | | | | | | |
| Zero-shot | 55.45 | 68.96 | 62.20 | 72.57 | 68.96 | 70.76 | 55.45 | 55.57 | 55.51 | 72.57 | 55.57 | 64.07 |
| Full FT | 71.73 | 90.28 | 81.01 | 111.22 | 90.13 | 100.68 | 71.67 | 86.68 | 79.18 | 110.86 | 87.46 | 99.16 |
| Ran Mask | 76.90 | 90.98 | 83.94 | 117.07 | 90.88 | 103.98 | 75.21 | 86.65 | 80.93 | 116.84 | 87.11 | 101.98 |
| Mag Mask | 77.89 | 90.78 | 84.34 | 117.35 | 91.27 | 104.31 | 76.13 | 86.97 | 81.55 | 116.93 | 86.91 | 101.92 |
| Grafting | 73.29 | 90.63 | 81.96 | 112.28 | 90.18 | 101.23 | 70.58 | 87.11 | 78.85 | 110.72 | 87.30 | 99.01 |
| L2-Reg | 73.92 | 90.68 | 82.30 | 111.5 | 90.48 | 100.99 | 71.73 | 87.06 | 79.40 | 110.76 | 87.35 | 99.06 |
| NRCA | 79.63 | 90.08 | 84.86 | 122.50 | 90.73 | 106.62 | 80.96 | 86.62 | 83.79 | 122.60 | 86.11 | 104.35 |
|  | + 24.18 | + 21.12 | + 22.66 | + 49.93 | + 21.77 | + 35.86 | + 25.51 | + 31.05 | + 28.28 | + 50.03 | + 30.54 | + 40.28 |
| *Fine-Tune with $B = 24$* | | | | | | | | | | | | |
| Zero-shot | 55.45 | 68.96 | 62.20 | 72.57 | 68.96 | 70.76 | 55.45 | 55.57 | 55.51 | 72.57 | 55.57 | 64.07 |
| Full FT | 71.57 | 90.68 | 81.13 | 110.31 | 90.48 | 100.40 | 71.63 | 87.08 | 79.35 | 109.72 | 86.49 | 98.11 |
| Ran Mask | 75.62 | 90.23 | 82.93 | 115.83 | 90.73 | 103.28 | 75.55 | 87.03 | 81.29 | 116.13 | 87.27 | 101.70 |
| Mag Mask | 75.16 | 90.63 | 82.90 | 115.35 | 90.68 | 103.02 | 74.27 | 87.27 | 80.77 | 115.10 | 87.16 | 101.13 |
| Grafting | 71.64 | 90.03 | 80.84 | 110.09 | 90.53 | 100.31 | 68.88 | 87.1 | 77.99 | 110.21 | 86.73 | 98.47 |
| L2-Reg | 71.31 | 90.08 | 80.70 | 109.81 | 90.38 | 100.10 | 69.75 | 87.38 | 78.57 | 109.65 | 86.78 | 98.22 |
| NRCA | 81.20 | 90.18 | 85.69 | 123.22 | 90.43 | 106.83 | 80.93 | 86.15 | 83.54 | 123.84 | 86.89 | 105.37 |
|  | + 25.75 | + 21.22 | + 23.49 | + 50.65 | + 21.47 | + 36.07 | + 25.48 | + 30.58 | + 28.03 | + 51.27 | + 31.32 | + 41.30 |

in Appendix C, the training loss for open-response tasks does not exhibit a progressive loss tendency but instead demonstrates a stepwise loss behavior. We hypothesize that this phenomenon is because MLLM tends to memorize each sample rather than explore the underlying distribution patterns, also observed in (Li et al., 2024b; Lu et al., 2024a). In contrast, it is challenging to map the same choice to different multi-modal inputs for fixed-choice settings. Consequently, MLLM is encouraged to analyze the multi-modal inputs and make informed decisions.

With recall of the empirical open-response task minimization, it would blindly increase the prediction confidence, $\mathbf{p}_t^{\mathbf{y}_t}$. Therefore, in our work, we aim to reduce the reliance of MLLM on language priors and instead emphasize the role of visual features in prediction behavior. Generally, we encourage noisy visual inputs to achieve consistent prediction confidence comparable to the normal branch, thereby mitigating learning shortcuts.

To formalize this, we denote $I$ and $\tilde{I}$ as the average prediction confidence over ground-truth tokens under clean and perturbed visual inputs (see Eq. (4)), respectively. Let $z_t$ and $\tilde{z}_t$ be the predicted logits at position $t$, and let $\sigma(\cdot)$ denote the softmax function. The NRCA regularization seeks to minimize the relative deviation:

$$\min \left| 1 - \frac{\tilde{I}}{I} \right| = \min \left| \frac{1}{T} \sum_{t=1}^{T} \frac{\sigma_{y_t}(z_t) - \sigma_{y_t}(\tilde{z}_t)}{I} \right|. \quad (8)$$

Since $I$ is detached from the computational graph, this simplifies to:

$$\min \left| \sum_{t=1}^{T} \sigma_{y_t}(z_t) - \sigma_{y_t}(\tilde{z}_t) \right|. \quad (9)$$

Assuming that $\sigma_{y_t}(z_t) \geq \sigma_{y_t}(\tilde{z}_t)$ holds (a natural assumption since clean inputs tend to yield higher confidence), the optimization objective becomes:

$$\min \sum_{t=1}^{T} \left( \sigma_{y_t}(z_t) - \sigma_{y_t}(\tilde{z}_t) \right), \quad (10)$$

which encourages the perturbed-input confidence $\sigma_{y_t}(\tilde{z}_t)$ to approach that of the clean input $\sigma_{y_t}(z_t)$. Since $\tilde{z}_t = g(\phi(f(\tilde{x}^v)), h_t)$, where $f$ is the visual encoder and $\phi$ and $g$ denote the projection and decoding modules, this objective drives the network to produce robust visual representations that are invariant to input perturbations. As a result, our model is encouraged to ground its predictions more faithfully in visual evidence rather than language priors, thereby mitigating overfitting by preventing overconfidence on individual training samples. We provide the full implementation and formulation in Sec. 3.

## D. Additional Experiments

We provide the experiment comparison with different training batch sizes of $B \in \{32, 24\}$ in Tab. 7. We conduct experiments on the VILA architectures on the four downstream datasets with the tuning epochs as $E = 3$. Compared with existing Multimodal Large Language Model methods, our method consistently demonstrates effectiveness.

