# OpenReview forum: "Be Confident: Uncovering Overfitting in MLLM Multi-Task Tuning"
_ICML.cc/2025/Conference — ICML 2025 poster_

### Official Review · Reviewer_XwCL · 2025-03-06

**Overall Recommendation:** 4

**Summary:**

This paper proposes Noise Resilient Confidence Alignment (NRCA) to reduce overfitting in open-response tasks during multi-task fine-tuning of Multimodal Large Language Models NRCA enhances performance on tasks like image captioning and visual question answering, outperforming traditional fine-tuning methods. Experiments and comparisons validate its effectiveness and compatibility with other optimization strategies.

**Claims And Evidence:**

The claims are well-supported by clear and convincing evidence.

**Essential References Not Discussed:**

Authors could add the discussion with the multi-task learning which shares the similar research objective with this paper.

**Experimental Designs Or Analyses:**

The experimental results are thorough and well-rounded, as the authors conduct a variety of experiments comparing different methods across multiple downstream tasks.

**Methods And Evaluation Criteria:**

The authors respectively measure the multi downstream tasks performance and further utilize the average performance to evaluate the overall performance. Authors conducts experiments on four downstream datasets into two task types: image captioning (open-response) and visual question-answering (fixed-choice) to formulate the multi-task setting.

**Other Comments Or Suggestions:**

The paper is written in a complicated manner with many key words such as open-response and fixed choice. Provide the clear definition would be better for paper readability.

**Other Strengths And Weaknesses:**

For strengths:
1.	Novel Problem Focus: Identifies the under - explored issue of open - response overfitting in MLLM multi - task tuning, which is crucial as MLLMs are increasingly used for diverse tasks.
2.	Effective Solution: Proposes NRCA, a method that uses Gaussian - perturbed visual inputs and confidence alignment. It effectively reduces overfitting and improves performance, as shown by experiments on multiple datasets and MLLM architectures.
3.	Comprehensive Testing: Conducts wide-ranging experiments with two MLLM architectures LLaVA and VILA and four downstream datasets across open-response and fixed-choice tasks. Ablation studies further analyze the method's components.
4.	Proposed method is relatively resource-efficient. It doesn't need to store pre - trained MLLM weights, reducing computational burden. Furthermore, the method's hyper-parameters work well across different MLLM models

For weakness:
1.	Application Limitation NRCA is designed for alleviate open-response overfitting and when there are only the fixed-choice tasks, its effectiveness is less obvious, restricting its universal application. It encourages the further discussion on limitation.
2.	Typos Errors. In Table 3, not retaining two decimal places. Please carefully check for grammar and markup issues
3.	The rationale for Eq.5 to detach the normal branch lacks the explanation. The authors should discuss the reason behind it.

**Questions For Authors:**

The authors should add more detailed limitation discussion and check the typo error. The rationale behind the proposed operation should add relative discussion.

**Relation To Broader Scientific Literature:**

This paper presents a simple yet effective approach to alleviate the overfitting issue of the open-response task during the multi-task tuning process. It addresses a novel research problem within the field of MLLM fine-tuning.

**Theoretical Claims:**

The authors analyze token prediction confidence in Eq.7 and argue that open-response tasks tend to reach overly confident predictions due to an overreliance on inherent memorization.

---

> ### Author Rebuttal · Authors · 2025-03-27
>
> Dear Reviewer XwCL:
>
> Thank you very much for your valuable comments and constructive feedback. Below, we carefully address each of your concerns point-by-point, providing detailed explanations and additional evidence to clarify our approach and validate its effectiveness.
>
> **Q1: Limitation Discussion** (Other Strengths And Weaknesses & Questions For Authors)
>
> A1: Open-response and fixed-choice tasks reflect two complementary aspects of Multimodal Large Language Model (MLLM) capabilities: the ability to generate free-form responses to open-ended questions, and the ability to select correct answers from predefined options. These two task types are widely recognized and are also included in the pre-training datasets construction. Solely tuning on fixed-choice tasks would degrade into the single task optimization and limits model application scope. In contrast, open-response tasks are crucial for unlocking MLLM ability to generate free-form expressions, which go beyond the limitations of traditional classification. We have provided a discussion in Page 5. We will provide a detailed limitation discussion in our revised manuscript. Thanks for your valuable suggestions!
>
> **Q2: Typos Errors** (Other Strengths And Weaknesses & Questions For Authors)
>
> A2: Thanks for your tips. We will fix the typos and check grammar and markup issues in our revised manuscript！
>
> **Q3: Rationale for Normal Branch Detach** (Other Strengths And Weaknesses & Questions For Authors)
>
> A3: In Eq. 5, we aim to enhance the visual behavioral effect to improve prediction robustness and reduce reliance on language priors. To achieve this, we encourage the noisy prediction confidence $\tilde{\mathcal{I}}$ to align with the normal prediction confidence $\mathcal{I}$, enforcing a one-way alignment constraint. Therefore, we detach the normal branch to prevent it from being influenced by the noisy branch during optimization. We will provide a detailed rationale discussion in our paper. Thanks.
>
> **Q4: Clear Keyword Definition** (Other Comments Or Suggestions)
>
> A4: Open-Response task means the task of generating a response to a given question such as image captioning dataset. Fixed-Choice task denotes the task of selecting one of several options from the candidate list, e.g., visual question-answering dataset. We will clarify the definition in our revised manuscript. Thanks for your suggestion!

---

### Official Review · Reviewer_SJh5 · 2025-03-10

**Overall Recommendation:** 4

**Summary:**

In this work, the author focuses on Multimodal Large Language Models tuning. Specifically, the authors propose a method called Noise Resilient Confidence Alignment (NRCA) that aims to alleviate the issue of overfitting, particularly in open-response tasks during multi-tasks tuning. approach emphasizes maintaining consistent prediction patterns in MLLMs by enhancing the integration of visual cues alongside language priors. Through experimental evaluations, they show that their method improves model robustness by aligning token prediction confidence towards the normal visual branch and reduces the over-reliance on language priors. The authors conduct extensive experiments on popular model architectures, validating the effectiveness of their approach across various multi-task downstream settings, including tasks like image captioning and visual question answering .

**Claims And Evidence:**

Yes. The claims and evidence are clear and convincing.

**Essential References Not Discussed:**

None.

**Experimental Designs Or Analyses:**

Yes. The authors use common evaluation metrics such as CIDER and Top-1 Accuracy, depending on the task. They perform experiments on popular Multimodal Large Language Models (LLaVA and VILA) using datasets like Flickr30k, COCO-Cap, ScienceQA, and IconQA.

**Methods And Evaluation Criteria:**

The NRCA method is designed to alleviate open-response overfitting during MLLM fine-tuning by encouraging prediction consistency between distorted and normal visual patterns. CIDER and Top-1 Accuracy are used to evaluate performance on downstream tasks. Comparisons with baseline methods and ablation studies further validate NRCA’s effectiveness.

**Other Comments Or Suggestions:**

Refer to weakness.

**Other Strengths And Weaknesses:**

Pros:

-	The paper introduces an innovative problem of overfitting in open-response tasks during multi-task fine-tuning, a gap not previously explored in existing research. This adds a valuable contribution to the field.
-	The paper includes a thorough set of experiments across a range of datasets and downstream tasks, such as image captioning and visual question answering, demonstrating the effectiveness of NRCA. The use of popular benchmark datasets like Flickr30k and COCO-Cap strengthens the reliability and generalizability of the results.
-	The NRCA method is a unique approach to controlling overfitting. By encouraging prediction consistency between distorted and normal visual patterns, NRCA offers a more interpretable mechanism for preventing overfitting, compared to traditional fine-tuning methods.
-	The paper includes a notation table and algorithm description in the supplementary material, making it easier for other researchers to reproduce the results and understand the underlying methodology.

Cons:

-	Conceptual Discussion: The authors should further claim the different for your methods with existing methods. Why they can not solve the problem mentioned in the paper to highlight the contribution.
-	The explanation for Figure 3 is unclear. Does it means your method alleviat the reliance on language prior?
-	Dataset Construction. As for the MLLM tasks, the authors should report the textual prompt for different tasks for better illustration.

**Questions For Authors:**

Please refer to Other Strengths And Weaknesses parts.

**Relation To Broader Scientific Literature:**

The author introduces an interesting problem that during multi-task tuning, open-response tasks exhibit overfitting behavior, a phenomenon that has not been thoroughly explored in existing research.  Proposed method consider encourage prediction consistency to alleviate the reliance on language prior.

**Theoretical Claims:**

N/A.

---

> ### Author Rebuttal · Authors · 2025-03-27
>
> Dear Reviewer SJh5:
>
> Thank you for your thoughtful review and for raising key concerns regarding our work. We aim to address your concerns in our detailed responses below, hoping to provide clarity and demonstrate the effectiveness of our proposed approach.
>
> **Q1: Conceptual Discussion** (Other Strengths And Weaknesses)
>
> A1: In the context of Multimodal Large Language Model (MLLM) tuning on downstream tasks, empirical evidence shows a tendency toward overfitting on the target distribution. This is primarily due to the mismatch between the model large-scale capacity and the limited size of domain-specific samples. Existing investigations either introduce stiff penalty regularization terms or design partial update masks. However, such approaches typically focus on single-task adaptation and apply uniform anti-overfitting objectives across all examples, without accounting for task-specific differences. Under multi-task settings, different task types often exhibit inconsistent fitting behaviors. Our analysis reveals that open-response tasks are particularly prone to overfitting, largely due to their heightened dependence on textual modalities. To address this, we propose a novel approach that mitigates overfitting on the target distribution by enforcing confidence alignment from noisy visual inputs to their corresponding clean visual patterns. We will provide a detailed conceptual explanation for the proposed approach in our revised manuscript to highlight our novelty. Thanks for your valuable suggestions!
>
> **Q2: Figure 3  Explanation** (Other Strengths And Weaknesses)
>
> A2: Figure 3 illustrates the performance under both normal and noisy visual input for different methods. Existing methods show minimal performance gap regarding visual quality, which reflects the restricted affect from the visual branch and presents a high reliance on the textual prior information to generate the corresponding output response. We will provide a detailed explanation for Figure 3 in our revised manuscript. Thank you for your helpful suggestion.
>
>
> **Q3: Dataset Construction** (Other Strengths And Weaknesses)
>
> A3: For fixed-choice tasks (ScienceQA and IconQA), we use the textual prompt: “Answer with the option’s letter from the given choices directly.” For open-response tasks (Flickr30k and COCO-Cap), we collect five manually written instructions and randomly sample one as the prompt for each caption, as follows:
> - "Describe the image as simply as possible with a sentence or phrase"
> - "Give a brief summary of what you see"
> - "Provide a short description of the image"
> - "Write a short description for the image"
> - "Briefly describe the content of the image"
>
> We will provide a clear description of the dataset construction procedure in our revised manuscript. We will release the corresponding training and test JSON files in the final version. Thank you for your valuable suggestions!

---

### Official Review · Reviewer_ELAr · 2025-03-12

**Overall Recommendation:** 3

**Summary:**

This paper introduces "Noise Resilient Confidence Alignment" (NRCA), a method to address overfitting in multi-task fine-tuning of Multimodal Large Language Models (MLLMs). The authors observe that while fine-tuning MLLMs on multiple tasks, performance on open-response tasks (like image captioning) degrades over time, whereas fixed-choice tasks remain stable. They attribute this to MLLMs over-relying on language priors rather than visual information. The proposed solution injects Gaussian noise into visual inputs and enforces alignment between prediction confidence on noisy and clean inputs, thereby enhancing visual representation in the model. Experiments across multiple benchmarks (Flickr30k, COCO-Cap, ScienceQA, IconQA) with different MLLM architectures (VILA, LLaVA) show NRCA consistently outperforms baseline methods and other fine-tuning approaches.

**Claims And Evidence:**

The paper's primary claim that open-response tasks degrade during multi-task fine-tuning is well-supported by empirical evidence in Figure 1, showing performance drops on Flickr30k while ScienceQA remains stable. The claim that NRCA alleviates this issue is supported by extensive experiments across different datasets, model architectures, and fine-tuning parameters.
However, the claim that language priors are the root cause of this degradation is less convincingly established. While the authors observe that models produce similar outputs for normal and noisy images (Figure 3), this could have other explanations besides language prior reliance. The connection between confidence alignment and reduced language prior dependency lacks a strong theoretical foundation.
The empirical results are strong and consistent across various settings

Update: the last part was clarified during the rebuttal period.

**Essential References Not Discussed:**

Some important references that might improve the paper's contextualization:

Literature on modality bias in multimodal models that could provide alternative explanations for their observations

Other recent work specifically on vision-language alignment during fine-tuning that could offer complementary insights on overfitting prevention

Update: the author responded during the rebuttal period and cited meaningful comparable literature in the field.

**Experimental Designs Or Analyses:**

The experimental design is sound and comprehensive. The authors test across:

Multiple MLLM architectures (VILA, LLaVA)
Different model sizes (3B, 7B)
Various fine-tuning epochs (E=3, E=5)
Different batch sizes (B=16, B=24, B=32)
Multiple benchmark datasets

The ablation studies isolate contributions of key components. The authors also demonstrate robustness to hyperparameter settings and analyze the method's resource efficiency compared to alternatives.

**Methods And Evaluation Criteria:**

The proposed method is well-motivated from the problem observation and appropriate for addressing multi-task fine-tuning challenges. The evaluation across multiple benchmarks like COCO-CAP and IconQA, and , with both open-response and fixed-choice tasks, provides a comprehensive assessment of the approach.

**Other Comments Or Suggestions:**

- Consider expanding the analysis of why random mask tuning performs so well
The paper would benefit from more explicit evaluation of visual attention patterns before and after applying NRCA

**Other Strengths And Weaknesses:**

Strengths:

Novel identification of the open-response overfitting problem in multi-task MLLM fine-tuning
Simple yet effective confidence alignment approach that's architecture-agnostic
Comprehensive empirical evaluation across diverse settings
Practical resource efficiency compared to alternative methods

Weaknesses:

The theoretical connection between language priors and open-response overfitting needs stronger justification
Surprising effectiveness of random mask tuning (which achieves competitive results) isn't thoroughly analyzed

**Questions For Authors:**

How do you explain the strong performance of random mask tuning as it relates to language prior issue with open-ended q-a geneartion? Does this suggest the problem might not be specifically about language priors but perhaps more general optimization challenges?

The gains on fixed-choice tasks are sometimes modest or even negative. Could you explain why enforcing confidence alignment might occasionally harm performance on these tasks?

**Relation To Broader Scientific Literature:**

The work builds effectively on existing literature in MLLM fine-tuning, particularly addressing overfitting challenges. The authors appropriately contextualize their work within related methods like parameter-efficient fine-tuning and regularization approaches.

**Theoretical Claims:**

The paper doesn't present formal theoretical proofs but offers explanations for why NRCA should work. The rationale connecting confidence alignment to improved visual representation is reasonable but would benefit from a more rigorous formulation.

Update: The author rebuttal addressed my concern.

---

> ### Author Rebuttal · Authors · 2025-03-29
>
> Dear Reviewer ELAr:
>
> We sincerely thank you for your valuable feedback and hope our responses adequately address your concerns and merit a score revision.
>
> **Q1: Theoretical Claims of NRCA and Its Connection to Visual Representation**
>
> A1: Denote $I$ and $\tilde{I}$ as the average confidence over ground-truth tokens under clean and perturbed visual inputs (see Eq 4). Let $\tilde{z}_t$ and $z_t$ be the predicted logits at position $t$, and let $\sigma(\cdot)$ denote the softmax function.
>
> The confidence alignment objective of $\mathcal{L}_{\text{NRCA}}$ is:
>
> $$
> \min \left| 1 - \frac{\tilde{I}}{I} \right|
> = \min \left| \frac{1}{T} \sum_{t=1}^T \frac{\sigma _{y_t}(z_t)-\sigma _{y_t}(\tilde{z}_t)}{I} \right|.
> $$
> Since $I$ is detached from the computational graph, the optimization reduces to:
>
> $$
> \min\left| \sum_{t=1}^T \sigma _{y_t}(z_t)-\sigma _{y_t}(\tilde{z}_t) \right|.
> $$
> Assuming $\sigma _{y_t}(z_t) \geq \sigma _{y_t}(\tilde{z}_t)$ for all $t$ (a natural assumption since clean inputs typically yield higher confidence), the objective becomes:
>
> $$
> \min \sum_{t=1}^T \left( \sigma _{y_t}(z_t)-\sigma _{y_t}(\tilde{z}_t) \right).
> $$
> This encourages the ground-truth token probabilities under perturbed inputs to approach those under clean inputs, i.e.
> $$
> \sigma _{y_t}(\tilde{z}_t) \to \sigma _{y_t}(z_t).
> $$
> Since $\tilde{z}_t=g(\phi(f(\tilde{x}_v)),h_t)$, the loss encourages the learnable modules ($\phi$ and $g$) to produce robust visual features that preserve confidence on ground-truth tokens under perturbations, thereby improving visual representation quality.
>
> **Q2: Theoretical Connection Between Language Priors and Open-Response**
>
> A2: We consider token-level prediction in autoregressive multimodal generation, where the model learns the probability of a target sequence $y=(y_1, \dots, y_T)$ given a visual input $x^v$ and textual prompt $x^t$. The objective is to minimize the negative log-likelihood $L_{\text{CE}}.$
>
> In open-response tasks such as image captioning (eg Flickr), the output is a multi-token sentence of length $T=k$, with autoregressive factorization:
> $$
> p(y \mid x^v, x^t)=\prod_{t=1}^k p(y_t \mid y_{\lt t}, x^v, x^t).
> $$
> Due to the richness and uniqueness of each output, the model may overfit by memorizing input-output mappings and relying on linguistic priors in $y_{\lt t}$ rather than visual cues.
>
> In contrast, fixed-choice tasks (e.g., IconQA) involve single-token outputs ($T=1$) from a small label set:
> $$
> p(y \mid x^v, x^t)=p(y_1 \mid x^v, x^t),
> $$
> where prefix conditioning is absent and language priors are less effective. The model must therefore rely more on visual input, reducing overfitting.
>
> This analysis complements Appendix C and the observations in Fig 6, further clarifying the connection between language priors and overfitting in open-ended tasks.
>
> **Q3: Essential References Discussion**
>
> A3: Prior work on **modality bias** shows that MLLMs often over-rely on textual priors, causing visually misaligned hallucinations[1]. Contrastive decoding compares token distributions across views but adds inference cost[2]. Our method aligns token-level confidence during fine-tuning to improve understanding.
>
>
> For **vision-language alignment**, existing methods either build diverse instruction data or use reinforcement learning with designed reward signals, both requiring costly annotation or tuning[3]. Our method avoids these overheads and integrates directly into standard fine-tuning.
>
> [1] Multi-modal hallucination control by visual information grounding,CVPR2024
>
> [2] Mitigating Object Hallucinations in LVLMs through Visual Contrastive Decoding,CVPR2024
>
> [3] Visual-RFT: Visual Reinforcement Fine-Tuning,arXiv2025
>
> **Q4: Random Mask Tuning Effectiveness Discussion**
>
> A4: Random Mask Tuning applies stochastic masks to parameters, injecting noise to reduce reliance on specific subnetworks and implicitly regularize by limiting parameter co-adaptation. While effective on smaller models (VILA3B), its performance drops on larger ones (LLaVA7B), as shown in Tab 4, suggesting its effect comes from constrained optimization rather than addressing language prior dominance. In contrast, our method perturbs visual inputs and aligns token-level confidence with the clean visual branch, enhancing visual grounding and ensuring consistent performance across model scales. We will include a detailed analysis in the manuscript.
>
> **Q5: Visual Attention Patterns**
>
> A5: Due to limited space, please refer to **Q2 in Response to Reviewer H4RG**.
>
> **Q6: Fixed-Choice Tasks Performance**
>
> A6: The phenomenon of modest gain in fix-choice task is also present in several counterparts designed to alleviate overfitting. Though some of them perform marginally better on fixed-choice tasks, they tend to incur trade-offs and underperform on open-response tasks. By enhancing visual robustness through confidence alignment, our method achieves a better trade-off and consistently surpasses all baselines in overall multi-task performance.

---

### Official Review · Reviewer_H4RG · 2025-03-17

**Overall Recommendation:** 3

**Summary:**

In the reviewed paper, the authors identify overfitting in open-response tasks as a significant challenge in multi-task multimodal large language model (MLLM) fine-tuning. They propose a novel method called NRCA, which aligns prediction confidences between noisy and normal visual inputs to improve visual representation learning. The paper presents comprehensive empirical evaluations that demonstrate the effectiveness of the NRCA approach in addressing the identified issue.

**Claims And Evidence:**

The following claims are not well supported

`We argue that leveraging MLLM to achieve multi-task specialization is a more efficient approach than the conventional
one-to-one fine-tuning paradigm.`

How to define the multi-task specialization? The original MLLM training itself is somehow a `multi-task specialization`. Is this method applicable to MLLM training itself?

**Essential References Not Discussed:**

No

**Experimental Designs Or Analyses:**

Partially.

Although the author asserts that NRCA addresses the issue of relying on the language prior, the supporting experiments and analysis unconvincing in demonstrating that the final improvement stems from resolving this problem. Including a visualization could aid readers in better understanding the proposed method and its effectiveness.

**Methods And Evaluation Criteria:**

Yes.

**Other Comments Or Suggestions:**

Table 2 is a bit confusing without checking main text.

**Other Strengths And Weaknesses:**

The paper discusses its limitations, which is commendable. It also provides the code, enhancing transparency and reproducibility. Additionally, the design of the NRCA loss function is cleverly crafted, showcasing an innovative approach.

The limitation comes from the additional forward/backward pass.

**Questions For Authors:**

Could you design experiments to confirm that the observed improvement genuinely results from addressing the issue of reliance on the language prior?

**Relation To Broader Scientific Literature:**

A common method for reducing hallucinations in multimodal large language models (MLLMs) involves adjusting the next-token logits contrastively based on the standard prediction distribution. This paper primarily distinguishes itself through its unique approach to loss design.

**Theoretical Claims:**

None

---

> ### Author Rebuttal · Authors · 2025-03-28
>
> Dear Reviewer H4RG:
>
> Thank you very much for your affirmation of our work, as well as the insightful concerns and questions you have raised. We have carefully considered each comment and provided responses.
>
> **Q1: Multi-Task Specialization Definition** (Claims And Evidence)
>
> A1: Multi-Task Specialization refers to fine-tuning a pre-trained model for a specific downstream application, which normally has a limited amount of data, compared with the pre-training data scale. For example, the authors of LLaVA claim that it involves $150k$ unique language-image instruction-following samples. However, the specialization application samples are much more limited, i.e., $10k$ for target task. Thus, tuning on the downstream distribution tends to lead to overfitting on the specialization task. And we point out that open-response tasks easily appear the overfitting behavior within the multi-task specialization. We attribute this degradation to language prior reliance and propose Noise Resilient Confidence Alignment to enhance the effect of visual cues in the token prediction. Therefore, our method is applicable to Multimodal Large Language Model (MLLM) training itself but benefits more in the specialization tuning process. We will provide  a detailed multi-task specialization definition in our revised manuscript. Thanks for your valuable suggestions!
>
> **Q2: Language Prior Reliance Visualization Analysis**  (Experimental Designs Or Analyses & Questions For Authors)
>
> We assume that for Multimodal Large Language Model (MLLM), input tokens include System Tokens, Prompt Tokens and Visual Tokens and appear different contribution for the prediction output. Thus, we utilize the attention map between the first output token and the input token under the noisy visual input to visualize the contribution of different input token.  Besides, recent works [1,2,3] have shown that the MLLM tend to extract object information from the image at the middle layers. Thus, we plot the all layers and middle layers attention allocation for the first output token in the following Table. The results reveal that our method allocates more attention weights on the visual cue. Besides, in Figure 3 (on Page 5), Full Fine-Tuning shows a limited performance gap between noisy and normal visual signals, revealing the reliance on language prior rather than understanding the visual information. Therefore, both the attention allocation analysis and performance comparison demonstrate that our method effectively reduces reliance on language priors and enhances the overall performance of the model.
>
> *Table: Attention allocation and performance comparison with relative methods. We conduct the evaluation on the Flickr30k with noisy visual input. We tuning on the Flickr30k+ScienceQA datasets based on LLaVA architecture.*
> |Metric |Full-FT | Rand Mask | Mag Mask | Grafting | L2-Reg | Ours |
> |----|---|---|---|---|---|---|
> | System   (All) | 0.6837 | 0.6966 | 0.6927 | 0.6673 | 0.6791 | 0.6677 |
> | Prompt   (All)  | 0.1229 | 0.1297 | 0.1280 | 0.1241 | 0.1181 | 0.1225 |
> | **Visual**  (All) | 0.1933 | 0.1737 | 0.1793 | 0.2084 | 0.2029 | **0.2098** |
> | System   (Mid)  | 0.5865 | 0.5874 | 0.5859 | 0.5726 | 0.5826 | 0.5574 |
> | Prompt   (Mid)  | 0.1733 | 0.1843 | 0.1832  |0.1664 | 0.1623 | 0.1576 |
> | **Visual**  (Mid)| 0.2402 | 0.2282 | 0.2309 | 0.2609 | 0.2552 | **0.2850**|
> | Accuracy| 62.25 |57.00 | 60.07|   60.26 | 59.78 | **67.89** |
>
> **Attributed to rebuttal format constraints, we will include attention map in our final version. Thanks for valuable suggestions!**
>
> [1] EAZY: Eliminating Hallucinations in LVLMs by Zeroing out Hallucinatory Image Tokens, arXiv, 2025
>
> [2] From Redundancy to Relevance: Information Flow in LVLMs Across Reasoning Tasks, NAACL, 2025
>
> [3] Cross-modal Information Flow in Multimodal Large Language Models, CVPR, 2025
>
> **Q3: Table Design** (Other Comments Or Suggestions)
>
> Thank you for your suggestion. We will reorganize the formulation of Table 2 to make it clear and easy to understand!

---

### Decision · Program_Chairs · 2025-05-01

**Decision:**

Accept (poster)

**Comment:**

This paper addresses the increasingly relevant issue of overfitting in open-response tasks during the multi-task fine-tuning of Multimodal Large Language Models (MLLMs). This work proposes the “Noise Resilient Confidence Alignment” (NRCA) method, which augments the visual branch through Gaussian perturbations and enforces token-level confidence alignment between noisy and normal visual inputs. The central claim is that this approach mitigates excessive reliance on language priors, thereby enhancing the robustness of MLLM predictions. Extensive experiments on diverse tasks (image captioning vs. visual question-answering) and multiple MLLM architectures provide empirical evidence supporting the method’s effectiveness. Based on the positive evaluations and the authors’ rebuttal addressing most concerns, the meta-review recommends acceptance of this submission.